# DART: Articulated Hand Model with Diverse Accessories and Rich Textures

**Daiheng Gao**[1*]    **Yuliang Xiu**[2*]    **Kailin Li**[3*]    **Lixin Yang**[3*]    **Feng Wang**[1]
**Peng Zhang**[1]    **Bang Zhang**[1]    **Cewu Lu**[3]    **Ping Tan**[4]
[1]Alibaba XR Lab    [2]Max Planck Institute for Intelligent Systems
[3]Shanghai Jiao Tong University    [4]Simon Fraser University
{daiheng.gdh,yunzong.wf,funtian.zp,zhangbang.zb}@alibaba-inc.com
yuliang.xiu@tuebingen.mpg.de
{kailinli,siriusyang,lucewu}@sjtu.edu.cn
pingtan@sfu.ca

## Abstract

Hand, the bearer of human productivity and intelligence, is receiving much attention due to the recent fever of digital twins. Among different hand morphable models, MANO has been widely used in vision and graphics community. However, MANO disregards textures and accessories, which largely limits its power to synthesize photorealistic hand data. In this paper, we extend MANO with **D**iverse **A**ccessories and **R**ich **T**extures, namely **DART**. DART is composed of 50 daily 3D accessories which varies in appearance and shape, and 325 hand-crafted 2D texture maps covers different kinds of blemishes or make-ups. Unity GUI is also provided to generate synthetic hand data with user-defined settings, e.g. pose, camera, background, lighting, texture, and accessory. Finally, we release **DARTset**, which contains large-scale (800K), high-fidelity synthetic hand images, paired with perfect-aligned 3D labels. Experiments demonstrate its superiority in diversity. As a complement to existing hand datasets, DARTset boosts the generalization in both hand pose estimation and mesh recovery tasks. Raw ingredients (textures, accessories), Unity GUI, source code and DARTset are publicly available at dart2022.github.io.

## 1   Introduction

Humans rely heavily on their hands to interact with surrounding objects and express their attitudes by sign language. Accurate reconstruction of these hand gestures from raw pixels, could facilitate the immersive experience in AR/VR, and lead us to a better understanding of human mental and physical activities. Emerging data-driven hand reconstruction approaches demand high-fidelity and diverse hand images, paired with perfect-aligned hand geometries. In addition to manually labeling the collected in-the-wild hand pictures, building large-scale synthetic data aided with photorealistic rendering engines and articulated hand model seems a promising and more affordable alternative.

However, existing articulated hand models [40, 36, 24, 23] are too idealized to represent the complexity and diversity of real hands. Realistic hands often vary in appearance (e.g. colors of skin and nails, palm prints), with blemishes (e.g. moles, scars, bandages), personalized make-up (e.g. tattoos), and accessories (e.g. ring, watch, bracelet, glove). Also, the captured textures with baked-in factors, i.e. lighting, shading, and materials, like HTML [36], are not suitable for photorealistic rendering pipeline. The comparison between different hand models is summarized in Tab. 1.

---

[*]These authors contributed equally to this work

36th Conference on Neural Information Processing Systems (NeurIPS 2022) Track on Datasets and Benchmarks.

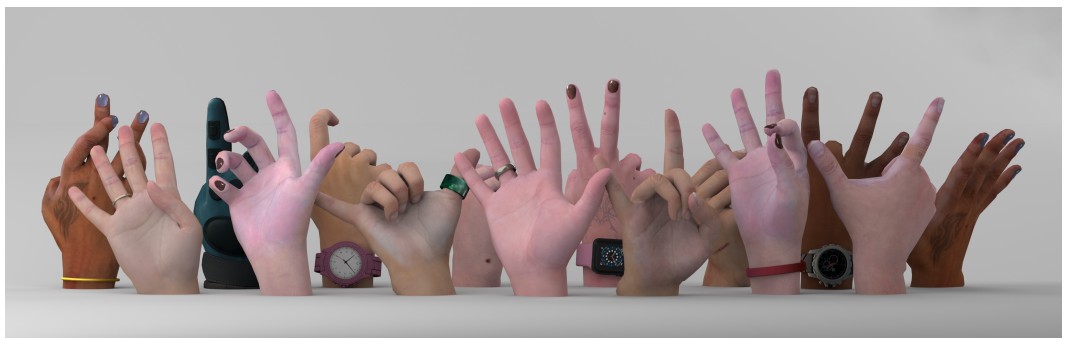

Figure 1: DART brings 3D hand model to a new level of realism. Aided with hundreds of high quality textures and additional accessories, photorealistic hand photos are synthesized.

Table 1: Comparison between different articulated hand models.

|  | MANO [40] | HTML [36] | NIMBIE [24] | DART (ours) |
|---|---|---|---|---|
| Skin Color | ✗ | ✓ | ✓ | ✓ |
| Albedo | ✗ | ✗ | ✓ | ✓ |
| Wrist | ✗ | ✗ | ✗ | ✓ |
| Muscle | ✗ | ✗ | ✓ | ✗ |
| Accessories | ✗ | ✗ | ✗ | ✓ |
| Num of 2D Textures | 0 | 51 | 38 | 325 |
| Num of 3D Accessories | 0 | 0 | 0 | 50 |

Therefore, neither **"Generalized Reconstruction"** — the hand estimator learned from synthetic images could generalize well to in-the-wild photos, nor the **"Friendly VR Setup"** — VR users do not need to take off their daily accessories, or, find a medical beauty clinic to remove their scars and tattoos before wearing VR Set, could be expected without a more realistic hand model with diverse accessories and rich textures.

To achieve this, we extend the MANO [40] as DART with the following novel features:

**a) Rich Texture:** Hand-craft UV albedo textures that adequately span diverse appearances (e.g. skin tones, nails, palm prints), together with blemishes (e.g. moles, scars, bandages), personalized make-up (e.g. tattoos), and daily accessories (e.g. ring, watch, bracelet, glove), see Fig. 1.

**b) Articulated Wrist:** DART is built upon wrist-enhanced MANO template, which could be driven by MANO's pose parameters, see Fig. 2b. Its importance lies on two-folds: 1) wrists always appear in real application, e.g. RGB(D) camera based hand tracking and reconstruction, but MANO was initially designed without it. 2) some daily accessories, e.g. watch and bracelet, are worn on the wrist.

**c) Diverse Accessories:** Daily accessories with both UV textures and 3D mesh, include different kinds of watches, rings, bracelets, and gloves, see Fig. 1.

Next, a synthetic data generator is constructed based on this DART model. Given the albedo texture maps, skin materials, lights, background photos, and the target pose distribution, Unity is used to render photorealistic images and export their paired hand pose and 3D/2D joint positions as well. The data generator has a GUI (see Fig. 3) comes with useful controllers, allowing users to carve hand images interactively. Also, the data generator can automatically render images based on a predefined setting. By this means, we create DARTset (see Fig. 6), a large-scale (800K) hand dataset with diverse poses, DART's exquisite textures and accessories. Each data sample in DARTset contains a photorealistic image and its corresponding MANO pose parameters, 2D/3D joints, mesh. All above software and data are available at `dart2022.github.io`.

DART could also be used to boost current hand pose estimation and mesh reconstruction tasks, we benchmark four representative reconstruction methods on DARTset. The quantitative results in Tab. 4 are well demonstrated that DARTset has great compatibility and generalizability. We also report the cross-dataset evaluations and justify that DARTset is a great complement to existing datasets.

## 2 Related work

### 2.1 Synthetic Hand Data

It is gradually realized by the computer vision community that, even though a neural network could somehow benefit from carefully designed layers, its performance is substantially restricted by the fidelity and diversity of training data. To further push the limit of data-driven approaches, people are starting to shift their focus from tedious manual labeling on collected photos to large-scale synthesizing using well-studied computer graphics and animation techniques.

*Synthesizing* has a few advantages over *manual labeling*: It can guarantee perfect and rich ground-truth labels with relatively low cost; users can control the diversity (e.g., pose, camera, background) w.r.t. requirements from specific users or scenarios; and it's easy and cheap to scale up.

Taking 3D human synthetic data as an example, AGORA [33], HSPACE [2] and GTA-Human [4] have proved their usefulness in downstream vision tasks, such as 3D pose estimation [9, 17], clothed human reconstruction [50] and human-scene interaction [5]. We won't discuss them in details since they are beyond the scope of this paper. Existing hand synthesizing methods can be grouped into three categories, summarized in Tab. 3

**1) GAN/VAE-based generation**: Based on CycleGAN [57], Mueller et al. [31] introduced GANer-ated Hands (GANH) to bridge the domain gap via syn2real image translation. GANH is a decent approach to resemble the distribution of real hand images. However, the authors don't make their articulated hand model publicly available, which limits its usage for other reconstruction tasks, e.g. mesh-based hand pose estimation, contact-aware hand-object interaction.

**2) Depth-based synthesis**: Wan et al. [48] propose Crossing Net, which models the statistical correlation of hand pose and its corresponding depth image by combining GANs and VAEs with a shared latent space. Oberweger et al. [32] introduces a hand depth video dataset with labeled 3D joints. Rogez et al. [38] synthesizes a hand-object depth data under egocentric workspaces. The model trained on these datasets can only be applied to the depth sensor's input, thus couldn't generalize well on hands wearing additional 3D accessories.

**3) Model-based rendering**: Zimmermann et al. [58] and Simon et al. [42] choose to synthesize hand data from Mixamo characters with a limited diversity of pose and skin colors. Rogez et al. [37] proposed an egocentric RGB-D video dataset rendered from commercial Poser [41]. SynthHands [30] is an RGB-D hand dataset, that is constructed by posing the articulated hand model with real mocap data, together with interaction and occlusion introduced by objects and clusters. Hasson et al. [13] presents a large-scale synthetic dataset of MANO hand grasping objects, called ObMan.

Though a few datasets already take skin tones into consideration, like GANH [31], RHD [58], and SynthHands [30], the additional accessories are missing. Basically, their hand proxy models are too *clean*. DART belongs to group 3), it introduces more diverse & complex textures and various 3D accessories, see Tab. 2, making a more sophisticated hand model.

### 2.2 Articulated Hand Models

Though MANO [40] provides the raw RGB scans used for registration, they are with baked-in textures. To decouple the albedo texture from raw RGB pixels is non-trivial. HTML [36] builds the hand texture model by compressing the variations of captured hand appearance to a low dimensional appearance basis using principal component analysis (PCA). But HTML still does not address the problem of baked-in lighting and shadow casting, and the hand appearance varies during articulation. Different from HTML's learned backed-in texture maps, DART provides diffuse maps disentangled from external factors, such as lighting and shading. Recent work NIMBLE [24] brings 3D hand model into a new level of realism, with bones, muscles and skins. However, none of above models consider daily accessories, which is the main contribution of DART. Besides, DART also adds common traits of hand inside the texture, like moles, nail colors, scars, tattoos and palm prints, see Tab. 2. And we propose a wrist-enhanced MANO hand tempalte, to synthesize hand data with wrist-based accessories, e.g. watch and bracelet, see Fig. 2b.

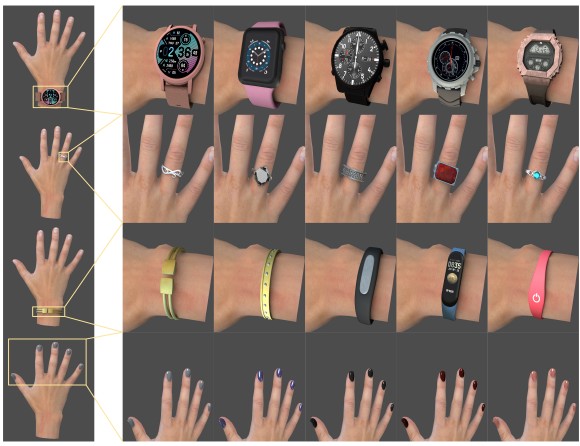

| Basic Texture | Amount |
|---|---|
| Bandage | $5 \times 3$ |
| Palm prints | $10 \times 3$ |
| Scars | $20 \times 3$ |
| Tattoos | $20 \times 3$ |
| Moles | $20 \times 3$ |
| Nail colors | $20 \times 3$ |
| Total | 285 |

| Accessories | Amount |
|---|---|
| Rings | 10 |
| Watches | 10 |
| Bracelet | 20 |
| Gloves | 10 |
| Total | 50 |

Table 2: Samples (left) and statistics (right) of DART's textures and accessories. All textures contain 3 basic skin tones: **dark, brown, light**. Note that since above skin tones are represented as the same 2D UV map, it's relatively easy to extend them from other skin tone libraries.

### 2.3 Image-based 3D Hand Pose / Mesh Reconstruction

Depending on the representation of the articulated geometry, 3D hand reconstruction can be categorized as Image-to-Pose (I2P) and Image-to-Shape (I2S).

I2P only focuses on the skeleton joints' locations of the articulation model. Existing I2P methods can be divided into two paradigms: heatmap-based [45, 34, 55] and regression-based [44, 39, 20]. Heatmap represents the 3D location of joints as Gaussian likelihood in a normalized 3D space. Regression-based methods map the input images to output joint locations. A representative method in each paradigm is Integral Poses [45] and Residual Log-likelihood Estimation (RLE) [20], respectively.

I2S then focuses on reconstructing full hand's surface geometry. The most common surface representation is the triangular mesh model (i.e. MANO [40]). MANO's vertices: $\mathbf{V} \in \mathbb{R}^{778 \times 3}$ are driven by the pose $\theta$ and shape $\beta$ parameters: $\mathbf{V} = \mathcal{M}(\theta, \beta)$, where $\mathcal{M}(\cdot)$ is a skinning function. Hence, the common practice in earlier I2S methods is regressing the $\theta$ and $\beta$ and to recover the hand mesh [54, 52, 3, 13]. Yet, the pose parameters are not defined in the Euclidean space (while the vertices are). The space shift hinders those methods from achieving higher performance. Later, several works [51, 56, 21] showed that the I2P can be integrated into I2S through neural inverse kinematics. These methods proved that I2P's accurate joints prediction facilitated I2S pose estimation.

Since mesh is a kind of graph, some works adopted graph-based convolution networks (GCN) to reconstruct hand. These methods leveraged the MANO's topology and used the spectral [18, 11] or spiral [19, 7] filtering to process the mesh vertices. GCN based methods achieved accurate reconstruction and are robust against abnormality. Recently, transformer-based [26, 25, 10] I2S methods have emerged. METRO [26] applying the self-attention on all the vertices-related features. It proved the superiority of involving non-local interactions among vertices. In addition to mesh-based hands, some I2S methods also seek to recover hand shape using other 3D representation, such as voxel [28], UV position map [6], and sign distance function [15]. In this paper, we benchmark four representative methods on DARTset, namely Integral Pose [45] (heatmap-based I2P), RLE [20] (regression-based I2P), CMR [7] (GCN-based I2S), and METRO [26] (transformer-based I2S).

## 3 DART data generation framework

Our framework is compatible with MANO's pose parameter and highly controllable. To achieve this, we decouple the texture maps, materials, ambient and point lights (position, intensity and color), backgrounds instead of using all-in-one baked-in texture maps. The narrative structure of DART data generation framework is as follows: Firstly, we detail how to create hundreds of exquisite texture maps and enhance the MANO's template hand mesh with wrist. Next, we describe DART's Unity GUI to generate photorealistic high-res images and its corresponding MANO pose and 2D/3D joints.

## 3.1 Texture map & Model

In the real scenario, 3D hand reconstruction and pose estimation tasks always take a hand crop as input, with former arm or wrist appearing in the image. To synthesize data with the similar structure with real-world input settings, we add a shaped wrist to the original MANO template hand mesh (778 vertices, 1,538 faces) as Fig. 2b shows. This composite structure, which contains 842 vertices and 1,666 faces, can be driven by the MANO pose $\theta_i \in SO(3), i \in 0, 1, ..., 15$ directly.

```
Input: W_init ∈ ℝ^{778×16}, P_init ∈ ℝ^{778×3×135}, R_init ∈ ℝ^{778×16}
1:  W_final = torch.zeros(842, 16)
2:  W_final[: 778] = W_init
3:  W_final[778 : 842] = W_init[777]
4:  P_final = torch.zeros(842, 3, 15)
5:  P_final[: 778, :, :] = P_init
6:  R_final = torch.zeros(842, 16)
7:  R_final[: 778] = R_init
Output: W_final, P_final, R_final, used for MANO's LBS
```

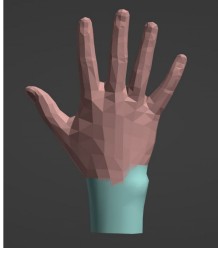

(a) Pseudo code of alignment process

(b) Wrist-enhanced MANO

Figure 2: DART hand parametric model.

To achieve this, we first remove the shape coefficients $\beta \in \mathbb{R}^{10}$ and only focus on finger articulation components, including blend weights, pose-dependent deformations, and joint-regressors. Next, we align the blend weights $W_{\text{init}}$, pose-dependent deformations $P_{\text{init}}$ and joint-regressors $R_{\text{init}}$ from 778 to 842, the extra 64 vertices are all numbered sequentially on the wrist while maintaining the palm and finger vertex number unchanged. The alignment process is shown in Fig. 2a

DART's 325 texture maps are designed and hand-crafted by five experienced 3D artists. As mentioned above, each texture map is of $4096 \times 4096$ resolution, some samples shown in Tab. 2. The creation of a texture map is as follows: we first create three basic texture maps in terms of skin tone: dark, light and brown. Then we add extra symbols, i.e, moles, nail colors, scars, tattoos and palm prints, or just fine-tune the basic texture map to get the various texture maps. Furthermore, we create dozens of high-quality 3D textured accessories, and place them on DART's template mesh. Given these hand meshes, we could render high-fidelity hand images, more details in Sec. 3.2.

The relative position of accessories on finger/wrist is FIXED to avoid collision. In this way, accessories on wrist, like bracelet and watch, could be transformed simply by applying root rotation. Regarding the rings on fingers, additional parent rotation is needed. Since the MANO's skeleton is represented in parent-child hierarchy, parent rotation could be easily computed along the kinematic tree.

## 3.2 Synthetic data generator

DART's another deliver is the synthetic data generator, based on `Unity3D`, allows us to render hands under controllable settings, e.g. poses, camera views, background, illumination (intensity, color, and position), and of course, DART's textures and accessories. Four main components are as follows:

**Lighting** We set two sidebars (ambient, directional) to control the position and intensity of lighting. Moreover, we add a palette for users who need to adjust the light color to mimic real-world scenarios.

**Controllable skeletal animation** Unity GUI supports skeletal animation for pose sampling. Given a hand motion sequence, users could adjust the speed, pause & export a specific pose frame.

**Pose refinement** As shown in the upper left panel of Fig. 3, DART enables users to fine-tune the position of bones manually. Hence, users could create a rare and challenging hand pose that is uncommon during automatic generation, to further improve its flexibility and diversity.

**Automatic data generation** For each selected or manually designed hand pose, firstly, the data generator randomly chooses a basic texture map and one background image. Among these subjects, 25% will be assigned a random accessory. Secondly, with all these ingredients, generator renders images under selected illumination and view. Please refer to `dart2022.github.io` for more details.

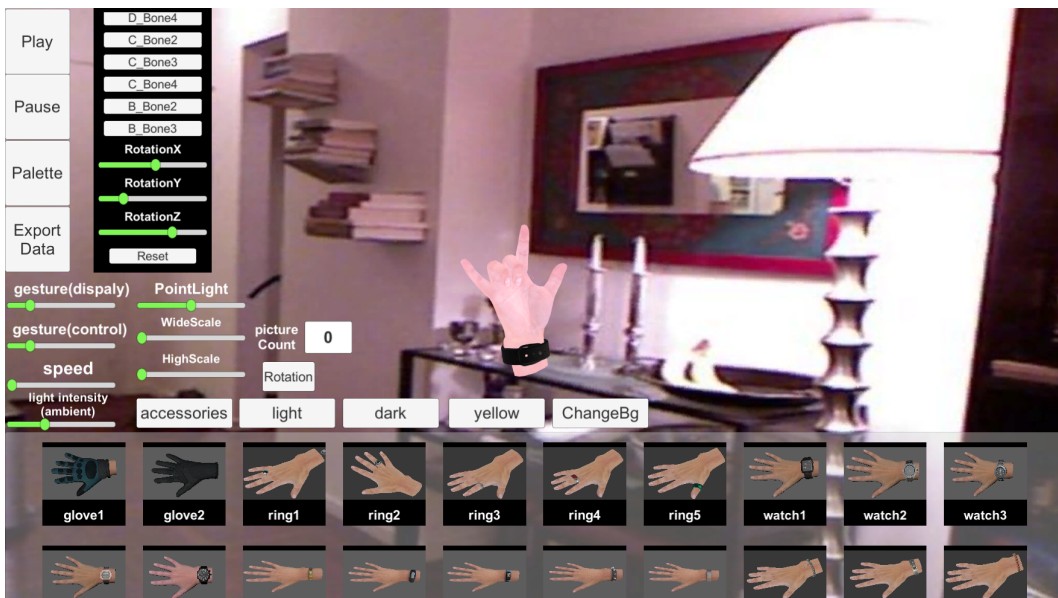

Figure 3: DART GUI for synthetic data generation. It supports adding textures, deforming skeleton, changing illuminations and backgrounds, and exporting the MANO poses.

Table 3: Comparison among RGB-based 3D hand datasets. Note that syn indicates synthetic data, real indicates real captured data, and Tex. & As. means the textures and accessories.

|            | STB [53] | RHD [58] | GANH. [31] | FreiH. [59] | ObMan [13] | InterH. [29] | DARTset (ours) |
|------------|----------|----------|------------|-------------|------------|--------------|----------------|
| Type       | real     | syn      | syn        | real        | syn        | real         | syn            |
| Size       | 36K      | 44K      | 331K       | 134K        | 153K       | 2.6M         | 800K           |
| Mesh       | ✗        | ✗        | ✗          | ✓           | ✓          | ✓            | ✓              |
| Tex. & As. | ✗        | ✗        | ✗          | ✗           | ✗          | ✗            | ✓              |

# 4 DARTset and Benchmark

## 4.1 DARTset

**Pose Articulation.** Pose articulation is a crucial step to augment pose distribution in DARTset. Hand's articulations are driven by one global wrist rotation and 15 fingers' relative rotations. To generate various global rotation, similar to MobRecon [8] and ArtiBoost [22], we uniformly adjust the viewpoints through sphere sampling. To generate various relative rotations, we discretize adequate poses within the hand's joint limits to cover all the possible configurations that a human could perform. We adopt the anatomically registered version of MANO: A-MANO [52] for conducting the discretion. A-MANO defines the legal rotation axes of each finger joint. By permuting all the legal discretized bending angles along the axes for each finger, we can get a group of the base poses.

However, these *clean yet fake* articulations are not diverse enough to approximate the real-world scenario. Hence, we introduce some *noise* from in-the-wild pose distribution, FreiHAND fits this requirement well. For each synthetic pose $\theta_i \in \mathbb{R}^{15 \times 3}$ from the aforementioned A-MANO, we first randomly choose 2,000 poses from FreiHAND, calculate the difference between $\theta_i$ and the 2,000, and select the one (denoted as $\tilde{\theta}_i$) that differs most from $\theta_i$. Then, we interpolate 8 rotations from $\theta_i$ to $\tilde{\theta}_i$ through spherical linear interpolation (Slerp) on quaternion. It is worth mentioning that interpolation between synthetic and real pose could reduce the pose domain gap of pose distribution between DARTset and the real-world hand captures, and selecting the most different pose to conduct interpolation promotes DARTset's pose diversity.

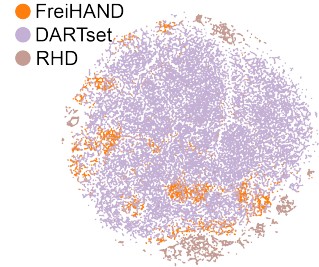

Figure 4: TSNE visualization for pose distribution of RHD, FreiHAND, and DARTset.

**Texture Composition.** We rendered the generated poses with random foreground texture map and accessories in DART, and with random background from COCO [27] dataset through alpha blending. As reported in Tab. 3, the total number of samples in DARTset is 800K. We split the DARTset into training, validation and testing set by the ratio of 0.8, 0.1, and 0.1. With the generator described in Sec. 3.2, we can easily expand the dataset to the number of billions. We project the hand pose in RHD [58], FreiHAND [59], and DARTset into the embedding space using t-SNE [47]. From Fig. 4, we conclude that compared with the synthetic data, RHD, DARTset has a closer distribution to the real-world dataset, FreiHAND. At the same time, DARTset has a more continuous and wider distribution than the FreiHAND dataset, which means our dataset has more

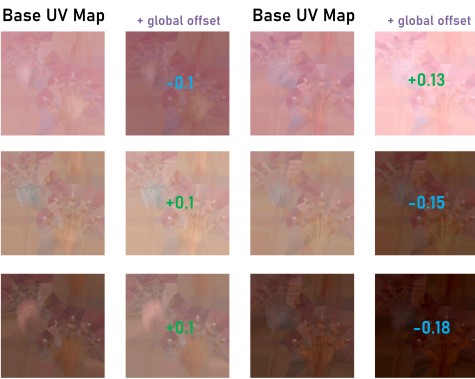

Figure 5: We augment the skin tones by adding global offset onto the basic UV textures.

generalizability. Besides, as Fig. 5 shows, we could added random global offsets $o \in [-0.15, +0.15]$ on top of three basic skin tones (dark, brown, light) to enhance their diversity. This simple augmentation operation could cover majority of human hand textures.

**Statistics on Data Generation.** DARTset is composed of train set (758,378) and test set (288,77). For every sampled hand pose, we randomly select a basic texture map together with a background image. Among these hands, 25% are assigned an accessory. Since accessory and texture map (skin tones, scars, moles, etc.) are uniformly sampled, the number of their renders are roughly equal. The resolution of rendered image is $512 \times 512$, and its corresponding annotations include 2D/3D joint positions, and MANO pose parameters. The whole process was executed sequentially, rendering process cost around 500ms per image on Windows11-empowered laptop with CPU (Intel(R) Core(TM) i7-10875H CPU @ 2.30GHz) and GPU (NVIDIA GeForce RTX 2070).

## 4.2 Task, Metrics, and Benchmark

We benchmark four mainstream hand reconstruction methods on DARTset testing set in Tab. 4, which are grouped into two categories: 1) Keypoint-based: Integral Pose [45], RLE [20]; 2) MANO-based: CMR [7], METRO [26]. These baselines could serve as a reference in 3D hand pose estimation / hand mesh reconstruction tasks.

We re-implement the above four methods to fit our dataset and training pipeline. We use ResNet [14] as the backbone of the first three networks, and HRNet [49] for METRO following the same practice in their paper. We train all the networks 100 epochs using Adam optimizer [16].

All the training images are cropped at $1.5\times$ the hand's bounding box and resized to the resolution of $224 \times 224$. The outputs of Integral Pose and RLE are the joints' UVD coordinates within a normalized 2.5D space. Later, we transform the UVD coordinates to 3D locations by a weakly-perspective camera model. The outputs of CMR and METRO are the vertices' 3D root relative coordinates.

To evaluate these methods, we report results using two standard metrics: PA-MPJPE and PA-MPVPE. PA indicates a 3D alignment with Procrustes analysis [12]. Mean-Per-Joint-Position-Error (MPJPE) and Mean-Per-Vertex-Position-Error (MPVPE) calculate the Euclidean distance between the ground truth and predicted results on hand's joints and vertices, respectively. To note, since the keypoint-based methods (Integral Pose and RLE) only infer joints' positions (without vertices), only the PA-MPJPE can be evaluated. For a fair comparison, although DART's hand has 842 vertices, PA-MPVPE only measures the distances to the 778 vertices of MANO.

Table 4: 3D hand pose / mesh reconstruction results on four learning-based methods.

|  | Integral Pose [45] | RLE [20] | CMR [7] | METRO [26] |
|---|---|---|---|---|
| PA-MPJPE ↓ $(cm)$ | **3.52** | 4.45 | 4.84 | 3.96 |
| PA-MPVPE ↓ $(cm)$ | - | - | **3.46** | 3.52 |

According to Tab. 4's col 1&2, Integral Pose outperforms RLE in terms of position errors. We offer a possible conjecture: the RLE models the deviations of the annotated keypoint position from its actual ground-truth. Since the rendered images in DARTset lack the inherited noise from the real-world capturing system, and the synthetic dataset lacks the uncertainty on its auto-generated annotations, RLE's normalizing flow will degenerate to a nearly identical transformation. Hence, the RLE model degrades to an ordinary regression model, which simply regressing the joints' positions. As for the two mesh-based networks (Tab. 4: col 3&4), METRO does not perform as well as CMR. We speculate this is caused by the METRO's transformer-based structure. METRO's attentions are conducted on all inputs tokens (vertices and joints queries), which is referred to as non-local interaction. Therefore, it may be less effective on capturing fine-grained local information. On the contrary, CMR leverages multi-level coarse-to-fine mesh structures and performs sequential spiral filtering based on those structures. Spiral filtering is able to improve the local interactions among neighboring vertices.

### 4.3  Ablation Study On Accessories

We conduct an ablation study to verify the effect of accessories. We use the same hand poses and camera views extracted from FreiHAND to re-render two datasets: DART with accessories and DART without accessories. Each dataset contains 32,560 images (same as the FreiHAND train set). We benchmark two learning-based models: Integral Pose and CMR on both datasets. As shown in Tab. 5, introducing accessories improves Integral Pose by 7.8% in terms of PA-MPJPE, and CMR network by 5.9% in PA-MPJPE and 7.2% in PA-MPVPE.

Table 5: Ablation study on training w/ and w/o DART's accessories.

| Method | Integral Pose [45] | | CMR [7] | |
|---|---|---|---|---|
| | w/o. Acs | w/. Acs | w/o. Acs | w/. Acs |
| PA-MPJPE $\downarrow$ ($cm$) | 6.15 | 5.67 | 7.65 | 7.20 |
| PA-MPVPE $\downarrow$ ($cm$) | - | - | 6.73 | 6.24 |

### 4.4  Cross-Dataset Evaluations

To demonstrate the merit of our DARTset, we report the cross-dataset evaluations on two mesh reconstruction methods: CMR [7] and METRO [26]. in Tab. 6 and Tab. 7. "Mixed" indicates we mix the FreiHAND dataset and DARTset equally in one batch to train the network.

We pick FreiHAND [59] as a representative dataset for three reasons: 1) FreiHAND is a field collected dataset with realistic lighting and environmental noise (compared to RHD, ObMan, and GANerated Hands). 2) FreiHAND has diverse camera views and hand poses (compared to STB). 3) FreiHAND is a commonly used benchmark. Instead, InterHand2.6M is a hand-interaction dataset. Half of the data has obvious self-interactions between hands. A domain gap still exists between our single-hand dataset DARTset and InterHand2.6M. Therefore, we only provide the baselines on the FreiHAND.

Table 6: Cross-dataset evaluation on CMR (PA-MPJPE / PA-MPVPE ($cm$)).

| test \ train | FreiHAND | DARTset | Mixed |
|---|---|---|---|
| FreiHAND | 7.41 / 7.50 | 25.64 / 25.73 | **6.70 / 6.83** |
| DARTset | 12.82 / 11.95 | 4.84 / 3.46 | 5.30 / 4.10 |

Table 7: Cross-dataset evaluation on METRO (PA-MPJPE / PA-MPVPE ($cm$)).

| test \ train | FreiHAND | DARTset | Mixed |
|---|---|---|---|
| FreiHAND | 7.35 / 6.94 | 17.73 / 17.58 | **6.88 / 6.85** |
| DARTset | 11.73 / 10.67 | 3.96 / 3.52 | **3.82** / 3.73 |

By comparing the column 1 and 3 in Tab. 6, we observe that the CMR model (Mixed training) improved 8.9% on PA-MPVPE on the FreiHAND testing set. This improvement beard out the fact that DARTset complements current challenging real-world dataset. However, column 2 and 3 reveals the domain gap between DARTset and FreiHAND, mainly in two aspects: 1) textures and accessories; 2) hand pose distribution, between FreiHAND and our DARTset. Since DARTset's large hand pose distribution dominates the gap (see Fig. 4), the mixed data training could greatly boost FreiHAND but not DARTset (wider pose distribution), which is the same case for METRO (in Tab. 7).

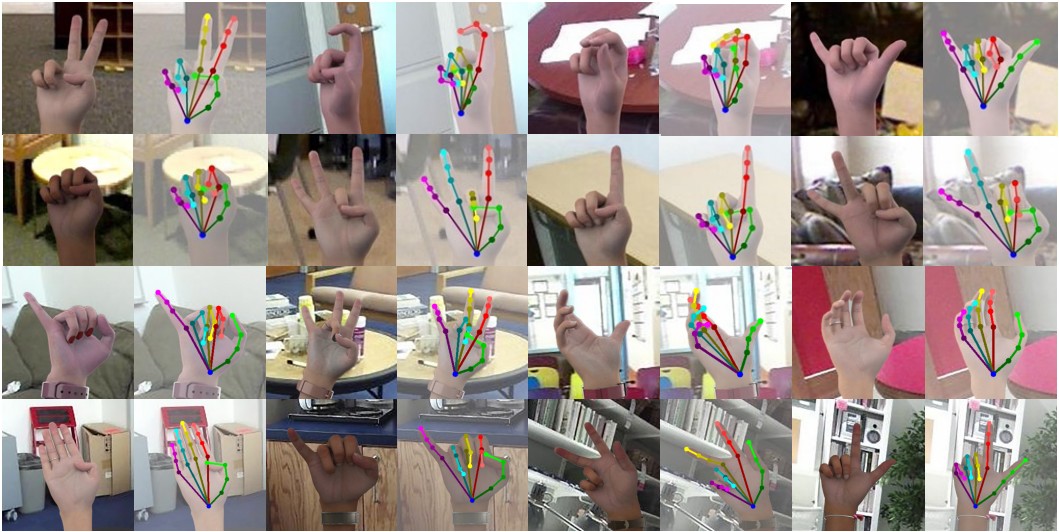

Figure 6: Diagram of DARTset. We show some rendered images and their 2D keypoints annotations. For better visualization quality, the displayed images are cropped around the center of hand.

## 5 Limitation and Future Works

This work mainly focus on generating hands with arbitrary gestures. The current pipeline is not optimized for random hand shape sampling. Thus, hand shapes remain fixed during the generation process. Besides, the 3D accessories are manually designed with fixed size, thus not adaptive to various hand shapes (e.g., watch, bracelet, gloves). How to add size-adaptive accessories in a fully automatic way is non-trivial. We leave this for future research. Also, DART currently does not support the hand-object or two-hands interactions, but since all the ingredients are publicly available and compatible with MANO, it's natural to extend DARTset with dynamic hand gestures, such as GRAB [46]. Last but not least, more advanced rendering techniques [43] or skin-specific shaders [35] (DART adopt mainstream SSS (sub surface scattering) skin shader [1] for now) could be utilized for more photorealistic rendering. We have released the full package on `dart2022.github.io` for only research purpose, including DART documentation, Unity executable package, source code, DART's texture maps, 3D textured accessories, and DARTset. All above will be available for a long time.

## 6 Acknowlegments

Yuliang Xiu is funded by the European Union's Horizon 2020 research and innovation programme under the Marie Skłodowska-Curie grant agreement No.860768 (CLIPE project). Kailin Li, Lixin Yang and Cewu Lu were supported by National Key Research and Development Project of China (No.2021ZD0110700), Shanghai Municipal Science and Technology Major Project (2021SHZDZX0102), Shanghai Qi Zhi Institute, and SHEITC (2018-RGZN-02046).

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
