# OpenReview forum: "DART: Articulated Hand Model with Diverse Accessories and Rich Textures"
_NeurIPS.cc/2022/Track/Datasets_and_Benchmarks — NeurIPS 2022 Datasets and Benchmarks _

### Official Review · Reviewer_qE3p · 2022-07-24
**A good pipeline to synthesize hand data**

**Rating:** 7
**Confidence:** 3
**Clarity:** The paper is easy to read and to follow.

**Strengths:**


+ Rendering hand data in CG software is a very good idea. Many comprehensive features of both hand and environment, i.e., textures, lighting and accesaries, are considered in the proposed dataset. Compared to existing hand datasets, the proposed one composites of larger amount of data and resemble real scenes more.

+ The synthesized data is shown to be useful in real cases. As shown in Table 5/6, training with the proposed dataset can improve the performance on real hand poses.

+ The authors claimed they would release the generation pipeline with a GUI. It is a big contribution in my view. With this GUI, one can collect realistic hand data easily and fast, which could burst relevant research.


**Weaknesses:**

- In this paper, the authors emphasized the importance of diversity of many factors (like textures and accessories). It is intuitively good and right. However, do these factors really have a big impact on relavant tasks? If possible, can the authors do some experiments to illustrate it? For example, to show the effect on hand color, the authors may train a model on light color hand and test it on both light and dark while keeping all other factors the same.
Meanwhile, it is not clear whether the accessories (rings, gloves ...) are counted into hand meshes, which needs further explaination

- This paper lacks detailed statistics on data distributions, e.g., hand shape (beta), background types, lighting parameters or RGB statistics..., of DARtset compared to existing datasets. Specifically, a comprehensive illustration or clear quantitative values on the *pose diversity* are not provided. Since hand poses shown in Figure 5 are in quite similar angles, it may lead the readers to worry whether all data in DARTset are like those. I think the authors could show more details on advantages of their dataset.

- Few things can be learned from the Benchmark part. Ideally, I think it should contain more analysis on performance and shortcomings of different methods instead of just a benchmark, if possible.

**Additional Feedback:**

If the GUI/dataset is released, and/or the authors provide more details, I will definitely improve the score.

**Correctness:**

The dataset is constructed in a sounding way, where hand data are rendered from Unity with correct settings.

**Documentation:**

This work provide a demo and some texture samples. It claims the full data and the GUI will be released in Aug.

**Ethics:**

I believe this work does not involve ethical issues like privacy and copyright since the data are rendered using CG software Unity.

**Relation To Prior Work:**

The paper contains a comprehensive related work section, but still need to show more details on difference of data distritbutions.

**Summary And Contributions:**

This paper propose a pipeline to synthesize hand data using the 3D rendering software Unity. To simulate human hands, many features like  color/texture/lighting/background/accessories are considered, which makes the data more realistic. With this pipeline, high quality hand data can be synthesized easily with ground truth labels. A large scale dataset named DARTset, consisting 800K hand images with associated rich labels are made within this pipeline. Remarkably, the GUI to make such data is claimed to be released to the public.

---

> ### Author Response · Authors · 2022-08-23
> **Address the concerns on benchmark, cross-dataset evaluation and open-source**
>
> ### Please check out [main paper](https://openreview.net/pdf?id=FPgCB_Z_0O) for our updates **(in sky-blue color)**
>
>
> **Q1: Fig4 needs to show more details on the difference in data distribution.**
>
> A1: Thanks. For each synthetic pose θi (15×3) from the A-MANO, we first randomly choose 2,000 poses from FreiHAND, calculate the difference between θi and the 2,000, and select the one (denoted as ˜θi) that differs most from θi. Then, we interpolate 8 rotations from θi to ˜θi through spherical linear interpolation (Slerp) on quaternion. For more details, please check Sec. 4.1. and Fig4 to get a better understanding of pose distribution difference.
>
> **Q2: The author claims the full data and the GUI will be released in Aug 2022.**
>
> A2: Sorry for the late reply, please check out the webpage https://dart2022.github.io for all those things. We have released the Unity executable package & source code in [GDrive](https://tinyurl.com/5n8bfcby), DARTset (train set, test set) in [Dropbox](https://tinyurl.com/2p875pa3), PyTorch Dataloader in [DARTset](https://github.com/DART2022/DARTset). For other documents and videos, please check out the webpage [https://dart2022.github.io](https://dart2022.github.io)
>
> **Q3: It is not clear whether the accessories (rings, gloves ...) are counted into hand meshes, which needs further explanation.**
>
> A3: Thank you for your suggestion. We conduct an ablation study to verify the effect of accessories. We use the same hand poses and camera views extracted from \freihand to generate two datasets: DART w & w/o accessories. Each dataset contains 32,560 images, same amount as FreiHand train set. We benchmark both datasets with two representative methods: **Integral Pose** and **CMR**. Introducing accessories improves Integral Pose by **7.8%** in MPJPE, and CMR network by **7.2%** in MPVPE. We have added such an ablation study table in Tab 5. Also, we have added more samples w accessories in Fig 6. In terms of the mesh reconstruction, we still only regress the 778 hand vertices for a fair comparison with FreiHand.
>
> **Q4: This paper lacks detailed statistics on data distributions. Since hand poses shown in Figure 5 (Now Figure 6 in our revision) are in quite similar angles, it may lead the readers to worry whether all data in DARTset are like those.**
>
> A4: Thanks for pointing this out. Fig. 4 has demonstrated that DARTset’s pose distribution is much more diverse than FreiHand. Please check [DARTset dropbox link](https://www.dropbox.com/sh/vugu0230u7767mw/AAA_D0qFZBN4F2wWyTRZyyDXa?dl=0) for a comprehensive understanding of pose & global rotation in DARTset.
>
> **Q5: Few things can be learned from the Benchmark part. Ideally, I think it should contain more analysis on performance and shortcomings of different methods instead of just a benchmark, if possible.**
>
> A5: Thanks. Our main insight here is to provide other researchers with the reference baselines for comparison, in 3D hand pose estimation & hand mesh reconstruction tasks. Listed four mainstream methods are grouped with regard to mechanism: 1) Keypoint-based: Integral Pose, RLE; 2) MANO-based: CMR, METRO.
>
> Have added this in Sec 4.2. We provide analysis here: During our experiments, we find that RLE is more unstable than Integral Pose. We offer a possible conjecture: Since the rendered images in DARTset lack the environment noise compared to the real image. The normalizing flow-based network, RLE, degrades to an ordinary regression network, i.e., simply regressing the joints' positions. As for the two networks that output mesh, METRO does not perform as well as CMR on vertices. We speculate this is due to the METRO's self-attention structure that ignores the local interactions between neighboring joints and vertices. While the graph convolution-based method, CMR, takes fine-grained local information into account. With the mesh topology prior, CMR achieves a better score on MPVPE.

---

> > ### Comment · Reviewer_qE3p · 2022-08-28
> > **Response to the authors**
> >
> > I appreciate the authors' effort on the response and the revision.
> >
> > My concerns have been largely addressed. Therefore, I have revised the score.
> >
> > However, there is a remaining issue. In Q3, I was concerned whether the accessories (rings/gloves) are regarded as a part of the hand mesh or not, *e.g.*, will the accessories affect the shape/size of the hand mesh?
> >
> > There are also some suggestions to further improve the work:
> > - The ablation study on accessories is good. But accessories are not the only contribution of the dataset. The author are expected to try more experiments on skin color/lighting/... to illustrate the advantage of your dataset.
> > - The diversity of the data is a contribution. The authors could illustrate it in an isolated chart/table with some quantitative values instead of a small visualization in the corner (Fig.4)
> > - The pseudo codes and the DART mesh between L.164 and 165 are in wrong format. It should be an isolated figure **with caption**.

---

> > > ### Author Response · Authors · 2022-08-28
> > > **Address the new concerns**
> > >
> > > We really appreciated your kind advice!
> > >
> > > For Q3, DART's accessories, e.g., rings, watches, bracelets,  and gloves, are **separate** textured mesh rather than part of the hand. Thus these accessories in DART won't affect the hand mesh. Please check [mesh plus accessories](https://github.com/DART2022/dart2022.github.io/blob/main/img/teaser_mesh.jpg?raw=true) for details.
> > >
> > > For diversity illustration, both Fig.1 and Fig. 6 have shown a bunch of qualitative samples of DARTset. Besides, Tab.1, 2, 3 already include some quantitative values. For other suggestions to further improve DART, we thank the reviewer for pointing this out and working on it to perfect the revision!

---

### Official Review · Reviewer_Pb7s · 2022-07-25
**A useful dataset and GUI for hands research with weaknesses in clarity and approach.**

**Rating:** 6
**Confidence:** 4

**Strengths:**

This is highly relevant work, as generation of synthetic hands data has received considerably less attention than other areas such as faces and rigid objects, while accurate collection and ground-truth of in-the-wild hands data is extremely difficult.  While it has weaknesses (described below), the dataset and GUI represent extremely useful data that does not have any available alternatives in the public domain. Hence the broader research community stands to gain significant benefit from the dataset.  Also, the use of artist-generated content avoids ethical concerns with collecting real-world data or even synthetic data based on real-world scans, hence this is a significant advantage. A few bullet points as well:

* The dataset as well as the raw materials for generating new data are diverse and appear reasonably high-quality.
* There is considerable diversity in textures, including wrist and accessories, which is fairly unique in the literature.
* The paper does a good job of describing the procedure for generating synthetic images and ground truth data.
* The method for choosing a set of hand poses in DARTset is clever, by pairing each “clean” hand pose with a maximally-distant pose in FreiHand, then using Slerp to interpolate several poses in-between them. The resulting diversity in pose appears quite high.



**Weaknesses:**

There are several weaknesses in clarity, approach, and features as detailed below. While the body of work that went into the work is significant, the communication is poor so may be difficult for other researchers to make good comparisons or learn a lot about how to create their own synthetic hands datasets. This might be mitigated by the contribution of assets and a GUI to generate data. However without seeing the code it is hard to tell if this is something that is easily extensible, or if the research community is likely to be able to easily adopt and extend it.


**Additional Feedback:**

There are areas of improvement that limit the usefulness of the data and approach. The current work does seem to be a good starting point to build off of and address these in the future however.
* Specifying and controlling dimensions of randomization:
    * For the GUI-based procedure, the randomization is only over texture and background, but doesn’t change the pose, illumination, or camera viewpoint. The latter would seem to be fairly straightforward, however as mentioned in “Limitations” section, randomization over pose would need some effort to do in a way that doesn’t create unrealistic poses so isn’t implemented here.
    * There did not appear to be any hand shape randomization (i.e., hand thickness, finger lengths, etc.)
    * Accessory placement is said to be randomized, but it is not clear how these are parameterized.  The difficulty go doing so is acknowledged in the Limitations section, but then what was actually done in the current paper?
* As mentioned in “Limitations”, there is no hand-object interaction.
* Only single-hand, not two-hand.
* No mention of accessibility concerns — e.g., missing fingers or injured hands.
* It is focused on generating single-frame poses, not animations, so no ability to use this yet for dynamic gesture recognition.

**Clarity:**

* In the description of the method for generating diverse poses in DARTset, it was not clear how a synthetic pose from A-MANO was chosen. Is there already a discrete set of realistic poses (not just realistic individual joints) provided in A-MANO?  The reference at this point is to a paper on “contact-potential-fields” for modeling hand-object interaction, which does not seem to be relevant here.
* The grammar could use significant help. The ideas are mostly clear enough that it was not impossible to understand the intent of the sentences, nevertheless the reading is a bit awkward.
* The code for GUI and asset generation are not discussed in detail so it is hard to tell if this work is easily extended


**Correctness:**

* There is only a bare baseline of using this data to train and test (1) a set of ML methods on the synthetic dataset, to show that models can train on this data and see how they compare with each other, and (2) comparing model accuracies when training either on synthetic-only, real-only, or a mixture.
* Any training involving real datasets is limited to FreiHAND.  It would be good to compare to other datasets such as InterHand2.6m.


**Documentation:**

No

**Relation To Prior Work:**

Yes

**Summary And Contributions:**

This paper describes a dataset of synthetic hands with diverse poses, textures, and backgrounds, for training hand-pose estimation models. It also promises to release the textures and accessories, and a Unity-based GUI, to generate synthetic hands data. The authors show that hand-pose inference models are able to train to high accuracy on a dataset of 0.8M synthetic-only images, and that training on a dataset with mixed real and synthetic data improves performance on real-only validation.

Note that there are really two contributions:
* (1)  A GUI, along with assets, to generate a batch of synthetic data for a given hand pose and lighting. The process of generating a target pose and lighting is somewhat tedious but then data is generated with randomized texture and backgrounds.
* (2) A dataset that uses bottom-up heuristics to sweep over a broad range of poses (based on A-MANO and FreiHAND), along with randomization over appearances.

---

> ### Author Response · Authors · 2022-08-23
> **Address the concerns on benchmark, cross-dataset evaluation and open-source**
>
> ### Please check out [main paper](https://openreview.net/pdf?id=FPgCB_Z_0O) for our updates **(in sky-blue color)**
>
>
>
> **Q1: Code or tools unavailable. The code for GUI and asset generation are not discussed in detail so it is hard to tell if this work is easily extended.**
>
> A1:  Sorry for the late reply, please check out the webpage https://dart2022.github.io for all those things.
>
> We have released the Unity executable package & source code in [GDrive](https://tinyurl.com/5n8bfcby), DARTset (train set, test set) in [Dropbox](https://tinyurl.com/2p875pa3), PyTorch Dataloader in [DARTset](https://github.com/DART2022/DARTset). For other documents and videos, please check out the webpage [https://dart2022.github.io](https://dart2022.github.io)
>
> **Q2: Any training involving real datasets is limited to FreiHand. It would be good to compare to other datasets such as InterHand2.6m.**
>
> A2: Thank you for your suggestion. We pick FreiHand as a representative dataset for three reasons: 1) FreiHand is a real dataset with realistic lighting and environmental noise (compared to RHD, ObMan, and GANerated Hands). 2) FreiHand has diverse camera views and hand poses (compared to STB). 3) Almost all the latest works benchmark on FreiHand.
>
> Instead, InterHand2.6m is a hand-interaction dataset with specific background and illumination. Half of the data has obvious self-interactions between hands. A large domain gap exists between our single-hand dataset DARTset and InterHand2.6m. Therefore, we only provide the baselines on FreiHand. We have updated this explanation in Sec 4.4.
>
> **Q3: Pose choices: how a synthetic pose from A-MANO was chosen. Is there already a discrete set of realistic poses (not just realistic individual joints) provided in A-MANO? The reference at this point is to a paper on “contact-potential-fields” for modeling hand-object
> interaction, which does not seem to be relevant here.**
>
> A3: Thanks for pointing this out. A-MANO defines the legal rotation axes of each finger joint. By permuting all the legal discretized bending angles along the axes for each finger, we can get a group of the base poses. MobRecon (CVPR2022) also used this method to build their dataset. We have clarified this in Sec 4.1.
>
> **Q4: Accessory placement.**
>
> A4:  In DART, the relative position of accessories on the finger/wrist is FIXED to avoid difficulty in dealing with collision issues. In this way, accessories on the wrist, like bracelets and watches, could be transformed simply by applying root rotation. Regarding the rings on fingers, additional parent rotation is needed. Since the MANO's skeleton is represented in the parent-child hierarchy, parent rotation could be easily computed along the kinematic tree. Really thankful for this advice, we have merged it into Sec 4.1.

---

### Official Review · Reviewer_GNor · 2022-07-26
**Paper review for "DART: Articulated Hand Model with Diverse Accessories and Rich Textures"**

**Rating:** 6
**Confidence:** 4

**Strengths:**

- Diversity in dataset in terms of appearance (skin color), blemishes, personalized make up, daily accessories
- Although not yet available, the tool would allow users to create hands interactively
- Supports a wide range of annotations photorealistic image and its corresponding pose parameters, 2D/3D joints, mesh, silhouette, and depth-map
- Also benchmarked DARTset on 4 representative methods

**Weaknesses:**

- Texture and accessories seems to be the distinguishing factor between this dataset and others (Table 3). Why is textures i.e. appearance, blemishes, make up and accessories so important? To better evaluate the importance of these features of generalisability, perhaps the authors could train on certain subsets i.e. no accessory vs accessory and evaluate them on other benchmarks.
- What is the effectiveness of training on DARTset? Perhaps authors could also train the same model on different datasets i.e. DARTset, FreiH, RHD, InterH. to evaluate the effectiveness of using DARTset over other datasets.
- For Figure 4, why was the distribution compared to RHD and FreiHAND and not the other 5 datasets mentioned in Table 3.
- The benefit of the articulated wrist allows additional wrist-accesories such as watch and bracelet. While this definitely increases diversity, I am not too sure how this will help hand reconstruction (PA-MPVPE and PA-MPJPE). Perhaps a subset with and without it could be run to evaluate the effectiveness.


**Additional Feedback:**

- The authors have yet to publicly release the tool and full dataset at this point of review.
- Would be a great feature to allow accessories to fit more variety in hand shapes

**Clarity:**

- Overall, well written and concise.
- Section 4.3 mentioned that CMR achieves 8.9% MPVPE. However, only PA-MPVPE were evaluated in Tables 5 and 6. Results on MPVPE could be added into the table if it helps to support the claim.
- Why is PA-MPVPE not added for Integral pose and RLE? If there's a reason, it could be added into section 4.2 for clarity.
- Why is PA-MPJPE  added in Table 4 but not Table 5 and 6. PA-MPJPE seems to be a useful metric to add in.
- Figure 4 is caption is slightly unclear. It is recommended to add the features are regarding hand pose distribution.

**Correctness:**

The dataset is constructed in a sound way. However, the effectiveness of training on this dataset is not evaluated or compared to other datasets. It would be interesting to compare how each diverse feature help to make it a strong dataset.

**Documentation:**

- The authors have yet to publicly release the tool and full dataset at this point of review.

**Relation To Prior Work:**

Table 1 and 3 lacks comparison to other datasets in terms of background, viewpoints, and pose diversity.

**Summary And Contributions:**

- Provided the Unity GUI which allows users to render hands with different pose, camera, background, lighting, and DART’s textures.
- Publicly released large-scale dataset DARTset containing a sizeable number of (800K) diverse hand images paired with aligned 3D annotations.
- DART extended MANO with textures and accessories

---

> ### Author Response · Authors · 2022-08-23
> **Address the concerns on accessory ablation, effectiveness of DARTset, code release, and variety in shapes**
>
> **Q1: no accessory vs accessory and evaluate them on other benchmarks. how wrist accessories will help hand reconstruction**
>
> A1: Thank you for your suggestion. We conduct an ablation study to verify the effect of accessories. We use the same hand poses and camera views extracted from FreiHAND to generate two datasets: DART w/ & w/o accessories. Each dataset contains 32,560 images, the same amount as FreiHAND train set. We benchmark both datasets with two representative methods: Integral Pose and CMR. Introducing accessories improves Integral Pose by 7.8% in MPJPE, and CMR network by 7.2% in MPVPE. We have added such an ablation study table in Tab. 5. Also, we have added more samples w/ accessories in Fig. 6. In terms of the mesh reconstruction, we still only regress the 778 hand vertices for a fair comparison with FreiHAND.
>
> **Q2: What is the effectiveness of training on DARTset? Perhaps authors could also train the same model on different datasets i.e. DARTset, FreiH, RHD, and InterH. to evaluate the effectiveness of using DARTset over other datasets. For Figure 4, why was the distribution compared to RHD and FreiHAND and not the other 5 datasets mentioned in Table 3?**
>
> A2: DART provides a series of toolsets, which enable generating hand images both *automatically* from existing pose libraries and  *manually* with the help of friendly Unity GUI, which is not available for other datasets. We pick FreiHAND as a representative dataset for three reasons: 1) FreiHAND is a real dataset with realistic lighting and environmental noise (compared to RHD, ObMan, and GANerated Hands). 2) FreiHAND has diverse camera views and hand poses (compared to STB). 3) Almost all the latest works benchmark on FreiHAND. Instead, InterHand2.6m is a hand-interaction dataset. Half of the data has obvious self-interactions between hands. A large domain gap exists between our single-hand dataset DARTset and InterHand2.6m. Therefore, we only provide the baselines on FreiHAND. We have updated this explanation in Sec. 4.4. As for the pose distribution comparison in Fig. 4, apart from the realistic dataset FreiHAND, to further demonstrate DARTset’s diversity and scalability, we choose RHD as the representative of the synthetic dataset.
>
> **Q3: The authors have yet to publicly release the tool and full dataset at this point of review.**
>
> A3: We have released the Unity executable package & source code in [GDrive](https://tinyurl.com/5n8bfcby), DARTset (train set, test set) in [Dropbox](https://tinyurl.com/2p875pa3), PyTorch Dataloader in [DARTset](https://github.com/DART2022/DARTset). For other documents and videos, please check out the webpage [https://dart2022.github.io](https://dart2022.github.io)
>
> **Q4: Would be a great feature to allow accessories to fit more variety in hand shapes.**
>
> A4: We have made a tentative attempt on circle-shape accessories, which could be uniformly scaled along the radius w.r.t. thickness of fingers or wrists. The attached texture maps could be reused since the scaling operation won’t change mesh topology. However, since the cross-sections of fingers and wrists are not perfect circles, it is non-trivial to avoid penetration. So we leave this for future research. This has been updated in Sec. 6.

---

> > ### Comment · Reviewer_GNor · 2022-08-26
> > **Several unaddressed concerns**
> >
> > Hi authors,
> >
> > Thank you for the updates. I appreciate the extra ablation studies on accessories and I think it helps to highlight the benefits.
> >
> > I have a few unaddressed comments under the "clarity" section:
> > - Section 4.2 mentions "With the mesh topology prior, CMR achieves a better MPVPE score". However,only PA-MPVPE results were given. Results on MPVPE should be added if it helps to support the claim, otherwise reporting it would cause confusion.
> > - Section 4.3 mentioned that with accessories, this improves "Integral Pose by 7.8% in MPJPE, and CMR network by 7.2% in MPVPE.". However, only PA-MPVPE metric were evaluated in Tables 5 and 6. Results on MPVPE and MPJPE should be added into the table if it helps to support the claim, otherwise reporting it would cause confusion.
> > - Why is PA-MPJPE added in Table 4 but not Table 6 and 7. PA-MPJPE seems to be a useful metric to add in.
> >
> > In addition, I have re-read the updated draft and I have a few more suggestions:
> > - I think it will be useful to provide MPVPE and MPJPE consistently as additional metrics.
> > - Section 4.2 is slightly unclear. "From above experiments, we find that RLE is more unstable than Integral Pose...". The analyses in this section references Table 4. However, Table 4 was not mentioned at all, which makes it confusing for the readers to look for the relevant results. You could start off with "Table 4 shows that comparison between XX methods...", or use [h!] to make sure the table lies within the Section.
> > - Section 4.3 should also reference Table 5. There's no mention on Table 5 in the text.
> > - Your claim of "in L285 We speculate this is caused by the METRO’s self-attention structure, which ignores the local interactions between neighboring joints and vertices. While the graph CNN-based method, CMR, takes fine-grained local information into account.".  I think this point can be elaborated more upon. In addition, I had the impression that METRO's models non-local iterations on top of local iterations.
> > - From Table 6 and 7. It seems that there exist a large domain gap between DartSet and FreiHand. For instance, training on FreiHand achieves PA-MPVPE errors of 11.95 on DARTset. However, training on DARTset achieves relatively high errors on FreiHand (25.73). This might raise concerns of whether this dataset will help generalisability to real-world use-cases.

---

> > > ### Author Response · Authors · 2022-08-27
> > > **Thanks for scrutinize DART!**
> > >
> > > First of all,
> > >
> > > We sincerely pay respect to you for helping us polish the experimental part of DART. Here are our replies:
> > >
> > >
> > >
> > > * **section 4.2 mentioned CMR achieves a better MPVPE score:**
> > >
> > > Sorry for the confusion. We report all the Mean Per-Joints/Vertex Position Errors with Procrustes Analysis. The reason for choosing Procrustes alignment is it focuses more on pose (https://files.is.tue.mpg.de/black/talks/SMPL-made-simple-FAQs.pdf). Hence here is the PA-MPVPE score. we have clarified this sentence in the revision.
> > >
> > >
> > >
> > > * **inconsistent PA-MPJPE & PA-MPVPE evaluation**
> > >
> > > For Tab.6 and 7, we have updated the PA-MPJPE score in the new revision Sec. 4.4
> > >
> > > As for Tab.5:
> > >
> > > Keypoints-based method (Integral Pose and RLE) only infer joints' positions (not vertices), hence only the PA-MPJPE can be evaluated.
> > >
> > >
> > >
> > > * **be useful to provide MPVPE and MPJPE consistently as additional metrics.**
> > >
> > > Following the same practice in CMR (CMR's table.1, 3, 5 ) and METRO,  we initially only report the Procrusted Aligned MPJPE and MPVPE.
> > >
> > >
> > >
> > > * **unclear reference in Sec 4.2 and 4.3**
> > >
> > > Thanks for your suggestion, we have updated the cross-paragraph page reference. please refer to Sec. 4.2.
> > >
> > >
> > >
> > > *  **Comparison of CMR and METRO can be elaborated more upon.**
> > >
> > > The METRO is a transformer-based regression network. METRO only has the encoder parts and all the attention is conducted on all inputs vertices and joints tokens, which is referred to as non-local interaction. Transformers are good at modeling long-range dependencies on the input tokens (in METRO, the input tokens are queries of joints and vertices), but they are less efﬁcient at capturing ﬁne-grained local information.
> > >
> > > On the contrary, the CMR leverages 4-level coarse-to-fine mesh structures and performs 4 times spatial filtering based on these topologies.  Spiral filtering (a type of Graph convolution) is able to improve the local interactions among neighboring vertices. We have updated the discussion in revision Sec. 4.2.
> > >
> > >
> > > * **It seems that there exists a large domain gap between DartSet and FreiHand**
> > >
> > >
> > > We attribute this to the domain gap, mainly in two aspects: 1) textures and accessories; 2) hand pose distribution, between FreiHAND and our DARTset. Since DARTset’s large hand pose distribution dominates the gap the mixed data training could greatly boost FreiHAND but not DARTset (wider pose distribution), which is the same case for METRO.
> > >
> > > Since the ultimate goal of DART is: 3D hand pose estimator's generalization ability could be really boosted through synthesizing numerous hand crops with ground truth under arbitrary viewpoints, gestures, backgrounds, and illuminations. At the data level, we believe DART is the most advanced solution in solving estimating 3D poses from monocular images because of the flexibility and easy-to-use property of GUI. Everyone can synthesize the data they want.
> > >
> > > ### Please check out [main paper](https://openreview.net/pdf?id=FPgCB_Z_0O) for our updates **(in sky-blue color)**

---

### Official Review · Reviewer_PcVq · 2022-07-26

**Rating:** 5
**Confidence:** 3

**Strengths:**

1. This paper presents a dataset of high-resolution texture maps for the MANO hand, which will be of interest to those already working with the MANO hand.

2. This paper also presents a data generation pipeline to render synthetic hands, which the authors use to generate a dataset which they claim can be scaled up to 3M+ images.

3. DARTset claims to cover a more diverse range of skin tones than prior datasets, which is important when training data-driven models with broader population groups in mind.


**Weaknesses:**

1. Adding hand accessories is one of the claimed contributions of the work, but it is unclear from the experiments how accessories impact results. DARTset examples in Figure 5 appear to show hands w/o any accessories.

2. Section 4.3, the paper only compares against FreiHAND as a baseline on mesh reconstruction. These experiments would benefit from additional comparisons to baselines, such as those in Table 3.

3. See Ethics section.

4. The clarity and polish of the writing could be greatly improved. See Clarity and Additional Feedback sections.

5. No code has been released yet. Even a small example demonstrating how a user may download DART and render a DART-textured hand model would be appreciated, and ease concerns of future code release.


**Additional Feedback:**

The paper says that DARTset may be scaled up to billions of images (L253). Please provide details on how long data generation takes, and how much disk space is required per sample (and/or in aggregate). Additionally, I would recommend that the authors avoid making this claim, since the current size of DARTset is 800k, and future work describes plans on scaling up to only 3M (L292).

Please provide details on how Figure 4 was generated.

Table 1 provides a comparison between different articulated hand models. I think a visual comparison would be helpful as well, to assess differences in realism.

I recommend avoiding self-congratulatory language, as in:
- “ingenious simplicity” (L167)
- “unique charm” (L191)
- “superior” / “superiority”
- “major innovation” (L101-102)

---

On clarity (this list is not comprehensive, and I would recommend the authors conduct a thorough editing pass):

L44: “The Unity”, grammatical error

L46: “with above the controllers”, grammatical error

L52-54: comma splice

L58: “the deep network”, grammatical error

L134: “we separating”, grammatical error

L243: what does “embrace” mean in this context?


**Clarity:**

The paper contains grammatical errors and awkward sentences throughout. See Additional Feedback.


**Correctness:**

L179-L182 describes how 3D artists were hired to create the texture maps. The paper would benefit from additional details on how the texture maps were designed, such as what these artists were instructed to do.


**Documentation:**

Google Drive links for the texture maps and DARTset are provided. However, code is not yet released (the authors say release will happen in Aug 2022).


**Ethics:**

Table 2 caption says that DART texture maps cover (black, light, brown) skin tone categories and (female, male) genders, for each type of basic texture and accessory. L99-100 says that DART can animate hands with “arbitrary skin tones”. The paper would benefit from additional discussion on how these base categories were selected, and how additional skin tones would be generated.


**Relation To Prior Work:**

Table 1 includes a comparison between relevant articulated hand models. The paper describes how this work differs in NIMBLE in L96-L103, namely:

1. high-resolution texture maps for the MANO hand model, rather than define a new parametric hand model as NIMBLE does

2. skin features such as “moles, nail colors, scars, tattoos and palm prints” (L101)

3. connecting forearm to wrist, for accessories such as watches and bands


**Summary And Contributions:**

This paper presents DART, which consists of 325 high-resolution texture maps and accessories for the MANO hand model. The paper also introduces DARTset, a dataset containing 800k images of hands generated using DART in diverse poses.

---

> ### Author Response · Authors · 2022-08-23
> **Address the concerns on ablation study, FreiHAND, code release, TSNE, generation cost, texture design, and overclaim**
>
> ### Please check out [main paper](https://openreview.net/pdf?id=FPgCB_Z_0O) for our updates **(in sky-blue color)**
>
> **Q1: It is unclear how accessories impact results. DARTset examples in Fig.6 appear to show hands w/o any accessories.**
>
> Q1: Thank you for your suggestion. We conduct an ablation study to verify the effect of accessories. We use the same hand poses and camera views extracted from FreiHAND to generate two datasets: DART w/ & w/o accessories. Each dataset contains 32560 images, the same amount as FreiHAND train set. We benchmark both datasets with two representative methods: Integral Pose and CMR. Introducing accessories improves Integral Pose by 7.8% in MPJPE, and CMR network by 7.2% in MPVPE. We have added such an ablation study table in Tab. 5. Also, we have added more samples w/ accessories in Fig. 6.
>
> **Q2: Section 4.3, why the paper only compares against FreiHAND as a baseline on mesh reconstruction**
>
> A2: We pick FreiHAND as a representative dataset for three reasons: 1) FreiHAND is a real dataset with realistic lighting and environmental noise (compared to RHD, ObMan, and GANerated Hands). 2) FreiHAND has diverse camera views and hand poses (compared to STB). 3) Almost all the latest works benchmark on FreiHAND dataset. Instead, InterHand2.6m is a hand-interaction dataset. Half of the data has obvious self-interactions between hands. A large domain gap exists between our single-hand dataset DARTset and InterHand2.6m. Therefore, we only provide the baselines on FreiHAND. We have updated this explanation in Sec. 4.4
>
> **Q3: No code has been released yet. Even a small example demonstrating how a user may download DART and render a DART-textured hand model would be appreciated, and ease concerns of future code releases.**
>
> A3: We have released the [Unity GUI & source code](https://tinyurl.com/5n8bfcby), [DARTset (train set, test set)](https://tinyurl.com/2p875pa3), [PyTorch Dataloader](https://github.com/DART2022/DARTset). For other documents and videos, please check out the [webpage](https://dart2022.github.io)
>
> **Q4: Please provide details on how Figure 4 was generated. Table 1 provides a comparison between different articulated hand models. I think a visual comparison would be helpful as well, to assess differences in realism.**
>
> A4: First, we randomly sampled 7% of the hand pose from each of the three datasets, RHD, FreiHAND, and DARTset, to get a tensor of N × 21 × 3. Then we reshape this tensor into an N × 48 matrices. The matrix is then projected into a 2D plane using TSNE [44]. Here we get an N ×2 matrix, indicating the coordinate on the embedding plane of each hand pose. Finally, the point from different datasets is labeled with different colors. Our DART’s hand skin texture is comparable to HTML’s scanned textures. Meanwhile, we provide exquisite texture mapping of scars, tattoos, accessories, etc. (shown in Fig. 1). Since the NIMBLE is not open source currently, we can not provide a comparison.
>
> **Q5: How long data generation takes, and how much disk space is required per sample (and/or in aggregate)**
>
> A5: Thank you for your inquiry about the details of data generation. The statistics are here: the final rendered image (w/o background) is of 512 × 512 resolution, ranging from 50 KB to 100 KB, the annotations file (include 2/3D joint position, mano pose parameter) is around 13 ∼ 15 MB per 1000 images. The whole process was executed sequentially, rendering process cost 500ms (on average) per image on Windows11 laptop with CPU (Intel(R) Core(TM) i7-10875H CPU @ 2.30GHz) and GPU (NVIDIA GeForce RTX 2070).
>
> **Q6: how the texture maps were designed, such as what these artists were instructed to do. The paper would benefit from an additional discussion on how these base categories were selected, and how additional skin tones would be generated.**
>
> A6: We consulted a professional art director for texture creation to confirm the core components of hand texture maps, i.e. resolution, skin color, fold, and palm print. We first create three representative skin tones: dark, light, and brown manually in MAYA on a watertight wrist-enhanced DART mesh. Then we recruited five 3D artists to enrich DART’s texture map database by making unique moles, nail colors, scars, and tattoos on top of the three basis skin tones, leading to 325 diverse texture maps in total. Moreover, we further augment the skin tones by randomly adding offset value o ∈ [−0.3, +0.3] to our basic texture maps (4096 × 4096 resolution). This has been added into Sec. 4.1, see Fig. 5 for additional skin tones
>
> **Q7: 3M dataset I would recommend that the authors avoid making this claim, since the current size of DARTset is 800k, and future work describes plans on scaling up to only 3M (L292).**
>
> A7: Thanks. We have already removed the 3M DARTset claim since it’s not been released yet on our [website](https://dart2022.github.io/).
>
>
> **Q8: Typos**
>
> A8: Thanks. we have updated some statements and fixed typos.

---

### Official Review · Reviewer_xPmt · 2022-07-27
**An important development for hands datasets but has some concerns**

**Rating:** 7
**Confidence:** 3

**Strengths:**

I believe that the proposed dataset and data generator are a valuable development for improving hand tracking and reconstruction. It is the first work to introduce skin imperfections and accessories to the dataset of hands, taking an additional step toward bringing the gap between synthetic and real data.

Providing a GUI to control the data generation process is another strong point — it would reduce the learning effort for the users of the generator. I’d also note that using a popular base hand model is also a good choice, which makes skills and tools for working with MANO transferrable to the new generator.

**Weaknesses:**

Firstly, my main concern is the potential for gender and skin tone bias in the data, which require clarification from the authors (I discuss this further in the Ethics part of the review).

Secondly, the dataset lacks standalone documentation requested in the NeurIPS datasets and benchmarks requirements.

Thirdly, the method needs a few clarifications that I request on in other parts of the review.

The first two are my main reasons for giving the ranking of 4.

**Additional Feedback:**

The authors chose the CC BY-NC-ND 4.0 license for their dataset, which includes the “No derivatives” condition. This clause may prohibit expanding the dataset, publishing new splits, etc., in the follow-up work. I ask the authors to confirm that they intend to restrict such derivative work.

**Clarity:**

The work delivers the intended message overall, but the explanations are unclear in several places, and some typos need fixing. Some of the things that I noticed are as follows:

- The sentence on lines 175-178 is hard to understand. I believe it should be separated into multiple, with each point providing more details.
- The sentence on lines 186-187 is strangely composed and refers to Table 1, which contains very different information.
- The following sentence (187-189) is also hard-to-read. Also, the word “stuff” is not typically used in academic writing.
- Lines 195-197. I could not understand the message at all. Do the authors provide the data loader? If yes, then what can it do? If that is an existing part of the pipeline, I ask the authors to introduce it in more detail.
- Line 219: “Exportation.” First of all, I believe it is “export.” On the other hand, judging by the provided description, it seems like something like “generation” would suit the procedure better. If I understood correctly, this feature allows generating several renders from the same pose, light setup, and camera view.
- The paper mentions “material properties” in several places. What is meant by those? How does it differ from the appearance provided by the hand textures?
- Sec. 4 of supplementary does not indicate that the licensing information is provided for the dataset and does not include licensing terms for the code and the assets (accessories, textures)

**Typos:**

Lines 90-91: “baked-in terms implied inside” → baked-in textures (?)

Line 139: detailed → detail

Line 145: is our GPU is (?)

Line 150: “which with former arm or at least wrist contained in the image”

Line 153: which of → “which contains” or “which consists of”

Line 165: below → above (I believe the mention was earlier)

Lines 183, 184: map → maps

Line 288: works → work

Line 293: acceptance → accepted

Supplementary, line 3: kinds of data → components (technically, GUI is not a data).

**Correctness:**

Overall, I feel that the suggested dataset and generator are constructed soundly. Extending the hand data with realistic textures and accessories is a natural direction to evolve existing synthetic datasets.

The cross-dataset comparison provides an overall demonstration that DARTSet complements FreiHAND well. However, the two datasets vary in multiple aspects — in pose distribution, skin tones (FreiHAND seems to be biased towards white skin), accessories, objects, and, likely, lighting conditions. It is hard to evaluate which aspects played a role in a performance boost. I would rather see a comparison that evaluated the work's main contributions, namely the texture details and accessories, and controlled for other factors, for example, by replicating the pose distribution of FreiHAND.

The way the authors handle accessories is not entirely clear to me. If I understood correctly, the shape of the hands could be modified using MANO-compatible shape parameters before starting the data generation. Since the shapes of accessories do not adapt to the hand shape change, how are the collisions handled? A more detailed description of the accessories' placement process is also needed. How are the allowed positions for accessories defined? These details are needed to understand the amount of variation in accessory placement and ensure the work's reproducibility.

As for the pose interpolation solution to enhance the pose distribution in DARTNet, I wonder how the authors ensured that interpolated poses are valid (e.g., do not contain self-collisions or unrealistic joint angles).

One the minor suggestions (do not affect the decision toward the paper):

- The GUI might benefit from utilizing Unity interfaces for pose manipulation instead of providing sliders for adjusting the pose.
- The naming conventions of the provided assets could be improved using distinctive features instead of numbers, e.g., glove_lace_smallpattern_white instead of lace_01_2. This approach will benefit the GUI and the overall usability of the assets.

**Documentation:**

The authors shared the details for data generation and maintenance information.

However, the authors did not provide separate stand-alone documentation for the created dataset using one of the frameworks recommended by the NeurIPS Datasets and Benchmarks track. One of the goals of the track is to promote best practices in creating, sharing, and maintaining datasets, and proper documentation plays a vital role in it.

The organizational structure of the dataset is not described, and the dataset sample is not provided (the authors only share the textures and accessory models), which prevents me from evaluating the dataset structure.

I would also like to see more statistics on the dataset beyond the total number of renders, poses, textures, and accessories. For example, how many renders are created per pose? Are an equal number of renders created for each accessory, skin tone, etc.? These questions are essential for understanding the distribution of hand appearances in the dataset.

Lastly, the authors did not mention the licenses on the assets (hand textures and accessory models) they created.

**Ethics:**

Overall, the dataset, being generated synthetically, does not pose concerns over subjects' privacy, but I see a potential for bias in sensitive categories.

I have concerns that the dataset may contain gender bias. The accessories introduced by the authors are grouped by gender, reflecting some of the typical gender biases (e.g., a women's watch has a pink strap, while a men's watch has a black strap, men's wristbands in gray colors vs. women's wristbands in yellow colors). If those differences are propagated to the dataset (e.g., "women" accessories only applied to women's hands and not men's), it creates a bias. The models trained on such a dataset may not be able to correctly process men's hands with brighter colored wristbands or women's hands with saturated colors of accessories. The authors did not specify the process of choosing accessories in relation to gender for the generated dataset, so I do not have the means to exclude the possibility of bias in the dataset.

Another concern is whether three base skin tones are enough to represent the skin tone diversity of humans. The dark skin tone is on the extreme side of the color spectrum, while the other two are on the lighter side. I am not entirely convinced that other shades of dark skin would be successfully processed by models trained on the provided dataset. SynthHands uses six skin tones, for example. Ideally, I would like the authors to either find the ground for the color selection in prior work or perform experiments that would justify the base color selection.

**Relation To Prior Work:**

The work stands apart from the prior art on synthetic hand datasets thanks to using accessories and skin imperfections, as indicated in the paper.

**Summary And Contributions:**

The work introduces a generator to create synthetic datasets of photorealistically rendered hands in various poses, camera angles, backgrounds, and lighting conditions. It extended a previously developed MANO hand, morphable model. The main novel points of the work are the introduction of handcrafted realistic textures that cover variations in skin tones, nail colors, tattoos, blemishes, and other surface variations, as well as providing a set of accessories of common types — watches, rings, bracelets, and gloves. The generator is supplemented with a GUI that allows fixing some of the settings giving more precise control over the data generation.

The paper also describes a dataset obtained with the developed simulator that provides more appearance variation compared to earlier work.

---

> ### Author Response · Authors · 2022-08-23
> **Address the concerns on ethics, documentation, license, detailed statistics, ablation study, valid pose, and accessories placement**
>
> First of all, many thanks for your detailed reviews, we have corrected the typos and updated the [main paper (in sky-blue color)](https://openreview.net/pdf?id=FPgCB_Z_0O) according to your feedback!
>
> **Q1: potential for gender and skin tone bias in the data.**
>
> A1: Thanks for mentioning this. Why we group the accessories by gender is to show their diversity, yet such gender information has never been leaked to the data generation pipeline, both texture maps and accessories in DARTset are randomly sampled (see Sec. 4.1) during synthesizing. Furthermore, MANO, the hand parametric model we extended with, is gender-agnostic. To avoid misinterpretation, we have updated Tab. 2 by removing gender categories.
>
> **Q2: whether three base skin tones are enough to represent the skin tone diversity of humans.**
>
> A2: Valuable suggestion. Though 3D accessory is our main selling point, since DART’s skin tones are represented in the same 2D UV space, it’s relatively easy to extend them from other sources by adding a global offset. Following the skin tones card in [Adobe Stock](https://stock.adobe.com/de/images/skin-tones-color-palette-vector/124986981), we could create more skin textures based on three basic skin tones, see Fig. 5.
>
> **Q3: lacks standalone documentation requested in the NeurIPS datasets and benchmarks requirements.**
>
> A3: For other documentation and videos, please check out the [website](https://dart2022.github.io)
>
> **Q4: CC BY-NC-ND 4.0 license includes the “No derivatives” condition. This clause may prohibit expanding the dataset, publishing new splits, etc., in the follow-up work.**
>
> A4: The raw assets (hand textures and accessory models), Unity package, source code as well as DARTset are now under Attribution-NonCommercial 4.0 International (CC BY-NC 4.0) instead of the previous CC BY-NC-ND 4.0 license. The best practice to create your own dataset is depicted on our [website](https://dart2022.github.io/).
>
> **Q5: statistics on the dataset beyond the total number of renders, poses, textures, and accessories. For example, how many renders are created per pose? Are an equal number of renders created for each accessory, skin tone, etc.?**
>
> A5: For every spherical interpolated hand pose, we randomly select a basic texture map together with one background image. Among these hands, 25% are assigned an accessory. In short, 1 render is created per pose. Since each accessory and texture map (skin tones, scars, moles, etc) are uniformly sampled, the number of their corresponding rendering images is roughly equal. We have added a detailed description of DARTset in Sec. 4.1.
>
> **Q6: It is hard to evaluate which aspects played a role in a performance boost. I would rather see a comparison that evaluated the work’s main contributions, namely the texture details and accessories, and controlled for other factors, for example, by replicating the pose distribution of FreiHAND .**
>
> A6: Thank you for your suggestion. We conduct an ablation study to verify the effect of accessories. We use the same hand poses and camera views extracted from FreiHAND to generate two datasets: DART w/ & w/o accessories. Each dataset contains 32,560 images, the same amount as FreiHAND train set. We benchmark both datasets with two representative methods: Integral Pose and CMR. Introducing accessories improves Integral Pose by 7.8% in MPJPE, and CMR network by 7.2% in MPVPE. We have added such an ablation study table in Tab. 5
>
> **Q7: I wonder how the authors ensured that interpolated poses are valid (e.g., do not contain self-collisions or unrealistic joint angles).**
>
> A7: The hand poses used for interpolation have two sources. First, similar to [5], we specify a group of legal base poses, i.e., permutations of legal discretize bending angles for each finger (based on A-MANO). Second, we adopt the legal hand pose from FreiHAND to introduce the noise from real-world distribution. Once we have guaranteed the legitimacy of the two sources poses, the question becomes how to make a valid interpolation between them. To this end, we adopt empirical evidence that spherical linear interpolation (Slerp) in the quaternion space from the legal base poses to the real-world poses is valid (similar evidence can also be found in the augmented dataset in [53]). Similar discussion in Sec. 4.1
>
> **Q8: Collisions between hand and accessories. A more detailed description of the accessories’ placement process is also needed.**
>
> A8: In DART, the relative position of accessories on the finger/wrist is FIXED to avoid difficulty in dealing with collision issues. In this way, accessories on the wrist, like bracelets and watches, could be transformed simply by applying root rotation. Regarding the rings on fingers, additional parent rotation is needed. Since the MANO’s skeleton is represented in the parent-child hierarchy, parent rotation could be easily computed along the kinematic tree. This has been merged into Sec. 4.1

---

> > ### Comment · Reviewer_xPmt · 2022-08-29
> > **Reply to authors**
> >
> > Thank you for promptly taking the feedback and improving your work and presentation!
> > The main points of my review were addressed. I've updated the score accordingly

---

### Official Review · Reviewer_UtnZ · 2022-07-27
**DART is an impressive synthethic hand dataset but some concerns are left for expriments**

**Rating:** 5
**Confidence:** 4
**Correctness:** Most claims are sound
**Clarity:** The paper is easy to follow.

**Strengths:**

The efforts for creating more realistic hand synthetic data is great.
1. Extend MANO model from 3 aspects, namely rich texture, diverse accessories and articulated wrist
2. Build a data generator to render synthetic hand data automatically
3. Perform experiments to verify the effectiveness of

**Weaknesses:**

Main concerns lie in the experiments.
1) Sec.4.2 benchmarks 4 representative methods, but it is not very clear the main insights.
2) Sec.4.3. performs cross-dataset evaluation, although reducing errors in mixed setting, some concerns are left for:

    2.1) From Tab.5. and Tab.6, the cross test seems to verify that a large domain gap exists between DART and FreiHAND

    2.2) It's unclear which factor contributes to the performance, and how much these different factors matters. Are these designed rich textures important? Or the designed hand poses are crucial? Or other factors like camera pose or rich accessories influence most.

    2.3) Although extending MANO with articulated wrist, it is unclear the importance of adding it from current expriments

**Additional Feedback:**

1. Will the data generator be made public?
2. I am willing to raise my rating if main concerns are addressed.


**Documentation:**

The data. ccollection procedure is clear. As 5 artists are recuited for creating textures, the authors may confirm that the made textures can be legally distributed.

**Ethics:**

No ethical concerns

**Relation To Prior Work:**

Most related works about synthethic hand dataset and articulated hand models have been discussed. Some synthetic human dataset, e.g., AGORA, HSPACE, GTA-Human, are also related. Although this paper focues on synthetic hand, synthetic human dataset performs similar pose/shape estimation task and have developed some techniques for using synthetic data.

**Summary And Contributions:**

DART is an extended-MANO with rich texture, diverse accessories and articulated wrist. A synthetic data generator based on Unity is built on top of DART to. generat 800K synthetic samples. With synthetic dataset DARTset, the paper demonstrates that it is ccomplement to FreiHAND.

---

> ### Author Response · Authors · 2022-08-23
> **Address the concerns on benchmark, cross-dataset evaluation, related works, data distribution, and open-source**
>
> Please check out [main paper](https://openreview.net/pdf?id=FPgCB_Z_0O) for our updates **(in sky-blue color)** .
>
> **Q1: Sec.4.2 benchmarks 4 representative methods, but it is not very clear the main insights.**
>
> A1: Our main insight here is to provide other researchers with the reference baselines for comparison, in 3D hand pose estimation & hand mesh reconstruction tasks. Listed four mainstream methods are grouped w.r.t. mechanism: 1) Keypoint-based: Integral Pose [42], RLE [ 17 ]; 2) MANO-based: CMR [6], METRO [ 23 ]. Have added this description in Sec. 4.2
>
> **Q2: Sec.4.3. performs cross-dataset evaluation, although reducing errors in a mixed setting, some concerns are left for:**
>
> - **Q2.1: From Tab.5. and Tab.6, the cross test seems to verify that a large domain gap exists between DART and FreiHAND**
>
> - A2.1: Good pointer. Domain gaps do exist between DART and FreiHAND, mainly in two aspects: 1) richer textures and accessories, and 2) hand pose distribution. Since the large hand pose distribution difference dominates the gap, see Fig. 4, DARTset covers a much wider pose distribution than FreiHAND, the data mixture could greatly boost FreiHAND but not DARTset. Moreover, we design the hand textures and accessories as realistic as possible. The domain gap could be further reduced with more advanced commercial renderers. We have added this in Sec. 4.4
>
> - **Q2.2: It's unclear which factor contributes to the performance, and how much these different factors matter. Are these designed rich textures important? Or the designed hand poses are crucial? Or other factors like camera pose or rich accessories influence most.**
>
> - A2.2: Thank you for your suggestion. We conduct an ablation study to verify the effect of accessories. We use the same hand poses and camera views extracted from FreiHAND to generate two datasets: DART w/ & w/o accessories. Each dataset contains 32,560 images, the same amount as the FreiHAND train set. We benchmark both datasets with two representative methods: Integral Pose and CMR. Introducing accessories improves Integral Pose by 7.8% in MPJPE, and CMR network by 7.2% in MPVPE. We have added such an ablation study table in Tab. 5
>
> - **Q2.3: Although extending MANO with articulated wrist, it is unclear the importance of adding it from current experiments**
>
> - A2.3: The importance of the articulated wrist has two folds: 1) wrists always appear in real applications, e.g. RGB(D) camera-based hand tracking and reconstruction, but MANO was initially designed without it. 2) some daily accessories, e.g. watch and bracelet, are worn on the wrist. This has been added into Sec. 1
>
> **Q3: Most related works about synthetic hand datasets and articulated hand models have been discussed. Some synthetic human datasets, e.g., AGORA, HSPACE, GTA-Human, are also related. Although this paper focuses on synthetic hand, synthetic human dataset performs similar pose/shape estimation task and have developed some techniques for using synthetic data.**
>
> A3: Good suggestion. We have added these synthetic human works in Sec. 2.1
>
> **Q4: The data collection procedure is clear. As 5 artists are recruited for creating textures, the authors may confirm that the made textures can be legally distributed.**
>
> A4: We have the exclusive property of DART’s textures since we have signed an exclusive contract with artists. We authorize the texture maps only for academic research purposes.
>
> **Q5: Will the data generator be made public?**
>
> A5: We have released the Unity executable package & source code in [GDrive](https://drive.google.com/file/d/1iqymPPPSF_rlKbHRvvgaVHlcmPsoEx25/view), DARTset (train set, test set) in [Dropbox](https://www.dropbox.com/sh/vugu0230u7767mw/AAA_D0qFZBN4F2wWyTRZyyDXa?dl=0), PyTorch Dataloader in [DARTset](https://github.com/DART2022/DARTset). For other documents and videos, please check out the webpage [https://dart2022.github.io](https://dart2022.github.io)

---

### Review · Ethics_Reviewer_VcHW · 2022-08-25

**Recommendation:** 1

**Ethics Documentation:**

Better documenting the limitations of the dataset in the limitations section of the paper can be helpful for users of the generator.

Also, having looked at the dataset website (https://dart2022.github.io/), there is no datasheet, which would be *very* useful for users of the generator. Refer to documentation from sources such as : https://raw.githubusercontent.com/huggingface/datasets/main/templates/README.md for a suggestion of what the datasheet should contain, and the Datasheets for Datasets article for further information: https://arxiv.org/abs/1803.09010

**Ethics Review:**

The ethical concerns raised by the reviewers are valid, and have been largely addressed in the revised version of the paper. The efforts made by authors around covering more skin tones and textures are definitely a great step forward.

I believe there is some simplification of the complexity of this task (representing the totality of human skin colors and textures in a realistic way) is a bit underestimated by authors, given the statement: "This simple augmentation operation could cover the majority of human hand textures." Referring to resources like the [Monk Skin Tone Scale](https://www.trustedreviews.com/explainer/what-is-the-monk-skin-tone-scale-4235590) can help ensure that this is truly the case.

---

> ### Author Response · Authors · 2022-08-26
> **Thanks for your kind suggestion and recommendation!**
>
> First of all, we are very grateful to you for pointing out our contributions and future endeavors (BRDF photorealistic skin rendering) in DART.
>
> For the datasheet or documentation, now we complete the **Dataset Organization** according to your useful reference(https://raw.githubusercontent.com/huggingface/datasets/main/templates/README.md) on our project page https://dart2022.github.io/.
>
> In a word, thanks for helping us in improving DART!

---

### Author Response · Authors · 2022-08-08
**Some links of DART**

Dear reviewers,

- Paper: [Latest version paper](https://openreview.net/pdf?id=FPgCB_Z_0O)
- Homepage: [dart2022.github.io](https://dart2022.github.io/)
- Unity executable package & source code: [GDrive](https://tinyurl.com/5n8bfcby)
- Dataloader for DARTset: [DARTset](https://github.com/DART2022/DARTset)
- DART's 335 texture maps & basic template hand mesh: [GDrive](https://tinyurl.com/ymf8kuxk)
- DARTset (train set + test set): [Dropbox](https://tinyurl.com/2p875pa3)
- Checklist: [GDrive](https://tinyurl.com/mr2a587y)

---

### Comment · Area_Chair_WcUK · 2022-08-25
**Reviewers: please respond to the authors**

Dear reviewers,

The authors have responded to your reviews. I encourage you to read and respond to them as well, before the discussion period ends.

---

### Author Response · Authors · 2022-08-27
**Ask for feedback from our rebuttals!**

Dear reviewers,

Thanks again for your insightful comments and constructive reviews that helped improve the paper! We are happy to see that most reviewers recognized the significance and value of DART and DARTset!

By now, we have
- 1. addressed your concerns in the replies
- 2. updated our [website](https://dart2022.github.io/) with more detailed documentation
- 3. updated the main paper **(in sky-blue color)** with some new descriptions, experiments, and explanations
- 4. released the [full package](https://openreview.net/forum?id=FPgCB_Z_0O&noteId=KXdYP_l03cx): full dataset (train set, test set), Unity GUI & code, PyTorch dataloader toolkit

Could you please take a look at them?

As there are **less than 24 hours** left in the discussion period, please let us know if parts of your concerns are still not addressed adequately! Otherwise, we would appreciate it if you translated our efforts into a revised score.

---

### Meta-Review · Area_Chair_WcUK · 2022-09-10

**Recommendation:** Accept
**Confidence:** 3

**Metareview:**

Four reviewers recommend acceptance, and two recommend rejection. Some reviewers, including an ethics reviewer VcHW, initially brought up issues with bias in skin tone, but they felt that these issues were largely addressed in the revision. Reviewer UtnZ recommended rejection. They raised concerns about the domain gap with FreiHAND, the analysis, and the utility of the articulated wrist. Reviewer xPmt initially recommended rejection based on concerns related to the cross-dataset comparison and handling of accessories, but recommended acceptance after the author response. Reviewer PcVq recommended rejection. They praised the diversity and size of the dataset, but had concerns about writing, claims around skin tone. They did not post a final review. Reviewer Pb7s recommends acceptance, based on the usefulness of the dataset and GUI, but had concerns about the writing it is unclear if these were addressed since there is no final review. Reviewer qE3p initially was concerned about diversity and analysis, but felt that these issues were addressed in the author response, and recommended acceptance. Reviewer GNor recommended acceptance, though had concerns initially about evaluation and about clarity. On balance, the AC recommends acceptance. However, the reviewers must address the concerns raised by the reviewers, including the issues around writing raised by PcVq (which are not completely addressed in the current revision) and the clarity issues raised by GNor.

---

### Decision · Program_Chairs · 2022-09-16

Accept